# Large Language Models as Source Planner for Personalized Knowledge-grounded Dialogues

**Hongru Wang**$^{\heartsuit}$, **Minda Hu**$^{\clubsuit}$, **Yang Deng**$^{\spadesuit *}$, **Rui Wang**$^{\diamondsuit}$, **Fei Mi**$^{\diamondsuit}$, **Weichao Wang**$^{\diamondsuit}$,
**Yasheng Wang**$^{\diamondsuit}$, **Wai-Chung Kwan**$^{\heartsuit}$, **Irwin King**$^{\clubsuit}$, **Kam-Fai Wong**$^{\heartsuit *}$

$^{\heartsuit}$Department of Systems Engineering and Engineering Management,
$^{\clubsuit}$Department of Computer Science and Engineering,
The Chinese University of Hong Kong
$^{\spadesuit}$National University of Singapore, $^{\diamondsuit}$Huawei Noah's Ark Lab
{hrwang, kfwong}@se.cuhk.edu.hk ydeng@nus.edu.sg

## Abstract

Open-domain dialogue system usually requires different sources of knowledge to generate more informative and evidential responses. However, existing knowledge-grounded dialogue systems either focus on a single knowledge source or overlook the dependency between multiple sources of knowledge, which may result in generating inconsistent or even paradoxical responses. To incorporate multiple knowledge sources and dependencies between them, we propose SAFARI, a novel framework that leverages the exceptional capabilities of large language models (LLMs) in planning, understanding, and incorporating under both supervised and unsupervised settings. Specifically, SAFARI decouples the knowledge grounding into multiple knowledge sources selection and response generation, which allows easy extension to various knowledge sources including the possibility of not using any sources. To study the problem, we construct a personalized knowledge-grounded dialogue dataset *Knowledge Behind Persona* (**KBP**), which is the first to consider the dependency between persona and implicit knowledge. Experimental results on the KBP dataset demonstrate that the SAFARI framework can effectively produce persona-consistent and knowledge-enhanced responses.

## 1 Introduction

Knowledge enhancement techniques (Yu et al., 2022) have significantly empowered machines to deepen their understanding of the underlying knowledge in open-domain dialogues (Huang et al., 2020), surpassing what can be solely acquired from conversational corpora. Recent years have witnessed various open-domain dialogue systems relying on different types of knowledge sources, such as topic (Wang et al., 2022; Zhu et al., 2023), external documents (*e.g.*, Wikipedia) (Dinan et al., 2019; Zhou et al., 2020; Wang et al., 2023c; Deng et al., 2023c), persona (Zhang et al., 2018; Liu et al., 2022a; Wang et al., 2023b; Deng et al., 2023b), user memory (Xu et al., 2022c,b; Park et al., 2023; Deng et al., 2022), and more. Realizing the limitations of using single-source knowledge, some recent studies further develop dialogue systems with access to multi-source knowledge (Wu et al., 2021, 2022b; Jang et al., 2022).

Despite their effectiveness of enriching the dialogue responses with multi-source knowledge, existing methods typically design models to incorporate all sources indiscriminately, resulting in a cumbersome process that struggles to handle cases dependent on the interaction between some specific sources instead of all (Wu et al., 2022b; Fu et al., 2022). Moreover, the importance of comprehending the potential dependency between knowledge sources is overlooked in previous works, which may result in generating paradoxical responses (Majumder et al., 2020; Jang et al., 2022). For example, humans often express their persona with the assistance of external knowledge. As shown in Figure 1(a), for responding to the question "*Hi, what do you like to eat?*", it is inadequate to only incorporate single-source knowledge from user persona, *e.g.*, "*I am a vegetarian*", since relevant information from external documents is also required, *e.g.*, *Vegetarian* (*Vegetarians like to eat fruits and vegetables*). However, being unaware of the dependency between these two different sources of knowledge (persona and documents), dialogue systems may select the document implying inconsistent personas (*e.g.*, "*Food contains meat, fruits, ...*"), leading to responses conflicting with defined personas (*e.g.*, "*I like to eat meat*"). Therefore, it attaches great importance in modeling the interaction and dependency of different sources[1] in building knowledge-grounded dialogue systems.

The absence of problem definitions and well-

---

$^{*}$Co-Corresponding Author.

[1]More dependency cases can be found in the Appendix A.

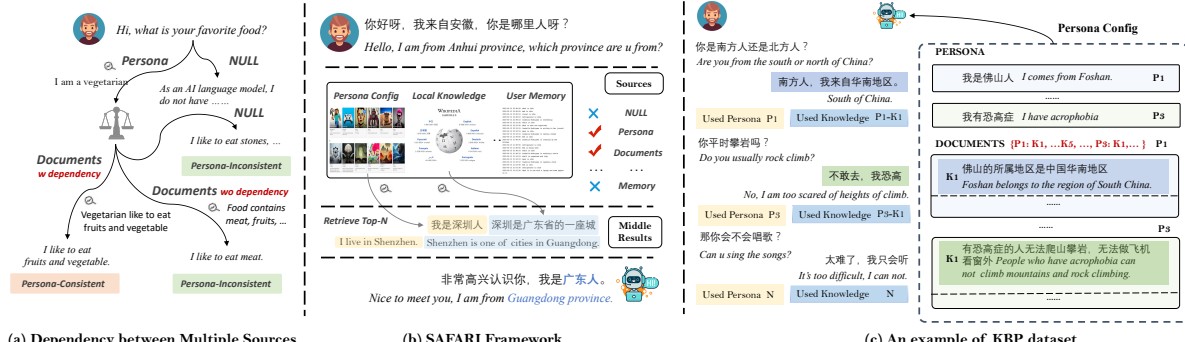

Figure 1: (a) An example of dependency of two sources involved in the persona-consistent dialogue system (PERSONA and DOCUMENTS); (b) our proposed SAFARI framework to plan, retrieve, and incorporate multiple sources of knowledge: PERSONA, DOCUMENTS, and so on. **Planning**, **Retrieval** and **Assembling** steps are divided by dashed lines; (c) A sample from the KBP dataset. There are three situations of responses in our datasets: 1) response without the need for any sources (NULL), 2) response using only personae description (from PERSONA source), and 3) response using both persona and knowledge (from PERSONA，DOCUMENTS sources). The example here presents the first and third situations. We highlight the response and used knowledge with the same color.

established benchmarks greatly impedes the progress of building dialogue systems that can capture the knowledge dependency between different sources. To this end, we construct a personalized knowledge-grounded dialogue dataset, named ***Knowledge Behind Persona*** (**KBP**), to mimic the scenario where the understanding of persona-knowledge dependency is required to produce consistent responses. KBP aims to build better persona-consistent dialogue systems with the help of utilizing underlying knowledge behind different persona descriptions comprehensively.

In order to address the interaction and dependency issue between specific sources, we propose a novel framework, named ***S**ource pl**A**nner **F**or person**A**lized knowledge-g**R**ounded d**I**alogues* (**SAFARI**). Seeing the potential of large language models (LLMs) in planning the use of external information (Schick et al., 2023; Shen et al., 2023; Gao et al., 2023), we explore LLMs' capability of connecting different sources in the context of the personalized knowledge-grounded dialogue system. As illustrated in Figure 1(b), the whole response generation can be modeled into three steps in SAFARI: 1) **Planning** to make a series of decisions of whether to use a specific knowledge source given relationship descriptions between different sources; 2) **Retrieval** to retrieve *top-n* results from external databases according to the decisions; 3) **Assembling** to incorporate all retrieved knowledge into the final response generation. Benefiting from decoupling source selection and response generation, our framework is more flexible and scalable, allowing independent modification of each component.

Additionally, our framework can easily accommodate scenarios where multiple or no sources are required. To sum up, our contributions are listed below:

- We propose the SAFARI framework to augment the dialogue system to plan and incorporate multiple sources of knowledge into responses ((e.g., decide whether or not require knowledge, which source to call, and when to call)), and further address the knowledge dependency issue between sources in both supervised and unsupervised manners by leveraging LLMs.

- We build a personalized knowledge-grounded dialogue dataset, KBP, where the responses are conditioned on multiple sources of knowledge, leading to more user-engaged dialogues with informative and persona-consistent knowledge.

- We conduct exhaustive experiments to validate the effectiveness of our proposed framework to incorporate multiple sources and capture the dependency between them[2].

## 2 Related Works

### 2.1 Personalized Dialogue System

To build a personalized dialog agent, Zhang et al. (2018) extensively investigated this task with a new dataset Persona-Chat, where a pre-defined persona set is a form of multiple sentences of textual description. Lots of works follow this setting and have taken mutual persona perception (Liu

---

[2]The code can be found in `https://github.com/ruleGreen/SAFARI/`.

et al., 2020; Kim et al., 2020; Xu et al., 2022a), persona-sparse scenario (Song et al., 2021; Welch et al., 2022), long-term persona memory (Xu et al., 2022c), persona extending (Liu et al., 2022b) and persona order bias (Chen et al., 2023) into consideration. Although some of them complement the insufficient semantics in short persona descriptions by further utilizing an external commonsense knowledge base to extend existing persona sets (Majumder et al., 2020; Liu et al., 2022b), they still fall into the conventional framework coupling the knowledge selection with the response generation (Wu et al., 2022b), rendering it infeasible to handle various sources of knowledge. There have also been works showing that the combination of different knowledge sources such as persona descriptions and Wikipedia can further improve the overall performance (Jang et al., 2022; Wu et al., 2021, 2022a). However, they still fail to capture possible dependency between knowledge sources. In their framework, knowledge is not used as the role to assist persona-consistent response generation, but as an additional resource to generate a more informative response (Dinan et al., 2019; Xue et al., 2023) or select a suitable persona (Jang et al., 2022; Fu et al., 2022).

## 2.2 LLMs for Planning

Large Language Models (LLMs) show remarkable capabilities in planning the use of various external resources, such as tools (Schick et al., 2023), models (Shen et al., 2023), and APIs (Li et al., 2023), to solve various NLP tasks and suit different applications in practice. Alternatively, different types of knowledge can be retrieved from external sources, as illustrated in WebGPT (Nakano et al., 2022) and ReAct (Yao et al., 2023). Integrating various knowledge sources to improve the quality of LLM generation becomes increasingly challenging due to the need for strategic planning, sequential decision-making, and complex reasoning. Previous research primarily focuses on either earlier decision-making stages (Nakano et al., 2022; Shen et al., 2023) or the subsequent response generation (Sun et al., 2023; Schick et al., 2023; Deng et al., 2023a), instead of establishing a complete framework for planning the use of multiple knowledge sources to generate appropriate responses. There is a latest work named TPE which regards different knowledge sources as *conceptual tools* and proposes a multi-persona collaboration framework to

model the decision-making process of the call order for multiple knowledge sources (Wang et al., 2023a). We differ in exploring the planning capability of LLMs to decide whether or not require knowledge, which source to call, and when to call in both the supervised and unsupervised manner.

## 3 Data Collection

In this section, we detailedly introduce the process of data collection and statistics of the collected data. The data collection process can be divided into two steps: **Step 1. Persona and Knowledge Acquisition** and **Step 2. Dialog Collection**.

### 3.1 Persona and Knowledge Acquisition

**Seeds Preparation.** To reduce annotation cost, we take advantage of the currently available persona dialogue dataset: DuLemon (Xu et al., 2022c) and two widely-adopted Chinese knowledge bases: Baike[3] and Ownthink[4] to produce seed data. Specifically, we first cluster all persona sentences from DuLeMon into 10 topics. After removing duplicate, rare, and similar personas, we carefully choose around 20 personas for each left topic as seed personas[5]. In addition, we manually add some personas for the existing topics and new topics. The final personas consist of *age, nation, personality, career, movie, music, sport, book, constellation, locality, gender, others*. For retrieving persona-related knowledge, we simply combine two aforementioned knowledge bases with similar filtering operations and store them as *(head entity, attribute, tail entity)* tuples.

**Persona and Knowledge Matching.** For each persona sentence, we segment it into a sequence of words with a Chinese word segmentation tool jieba[6]. If any words exactly match the head entity or tail entity of a certain knowledge tuple, we transform the tuple into a sentence according to predefined templates and then save it as one knowledge for this persona sentence. In this way, we can obtain various knowledge for each persona sentence. Consistent with previous works (Zhang et al., 2018; Majumder et al., 2020), we randomly sample 3 persona sentences along with 5 knowledge sentences per persona to form a persona description

---

[3]http://www.openkg.cn

[4]http://github.com/ownthink/KnowledgeGraphData

[5]Some topics are removed if they contain less than 20 personas. We don't pick hundreds of personas because one persona sentence has a vast amount of knowledge behind it.

[6]https://github.com/fxsjy/jieba

of the system for each dialog. More details and templates can be found in Appendix B.

## 3.2 Dialogue Collection

**Collection Setting.** Following the setting of Jang et al. (2022), annotators are instructed to make a dialogue by considering persona and corresponding knowledge under the single-person setup. In this way, one person can better understand what persona to ask as the human and what knowledge to use as the system, in comparison with two independent persons playing two roles separately. During the collection, each annotator first selects a suitable persona and then optionally identifies relevant knowledge, giving a knowledge-enhanced and persona-consistent response at last.

**Training and Pilot Annotation.** All annotators are first required to take a training tutorial to learn the annotation procedure, requirements, and examples of annotated dialogues. Afterward, they are given 30 personas to make 10 dialogues. We provide corresponding feedback to help them adjust their annotation criteria. To establish the necessity of persona-knowledge dependency, we consider the situation where the response will be persona-inconsistent without the assistance of knowledge. To this end, annotators are requested to ask questions centered on the implications based on knowledge and persona. For example, the annotator is supposed to ask *"Which province are you from?"* instead of *"Which province does Shenzhen belong to?"*, given the persona *"I live in Shenzhen"* and corresponding knowledge *"Shenzhen belongs to Guangdong province"*.

**Batch Collection.** After pilot annotation, we conduct dialogue collection batch by batch and regularly coach the quality of collected data[7]. For each batch, we sample personas different from previously annotated dialogues to increase its diversity in the whole dataset. The addition, deletion, and revision of persona and knowledge are also accepted and updated at the batch level[8].

**Grounding Annotation.** We also gather the labels of grounding knowledge sources for the system's responses by asking the annotators to specify the sources they draw from while providing responses, such as PERSONA or DOCUMENTS. For instance, gen-

---

[7]We write a python script to check typos (*e.g.* the labels of used knowledge is not exist in given knowledge bases) and provided feedback after each batch.

[8]The annotators must check for persona conflicts and refrain from relying too much on single knowledge.

| KBP | Train | Valid | Test |
|---|---|---|---|
| # dialogues | 1,981 | 248 | 248 |
| # samples | 9,821 | 1,227 | 1,229 |
| # avg turns | 4.96 | 4.93 | 4.96 |
| # utterances | 19,642 | 2,454 | 2,458 |
| # avg length | 17.6 | 17.3 | 17.5 |
| # resp w/ persona | 86.1% | 84.4% | 85.3% |
| # resp w/ p_and_k | 76.3% | 74.2% | 75.1% |

Table 1: Statistics of KBP dataset.

erating a response may rely on persona alone or both persona and knowledge. With the labels of these grounded sources, the planning abilities of the dialogue systems can be quantitatively measured.

## 3.3 Statistical Analysis

We organize the collected personas (PERSONA source), persona-related knowledge [9] (DOCUMENTS source), and dialogues in the form shown in Figure 1(c). We finally collect 2,477 dialogues and 24,554 utterances with 5 turns per dialogue on average. We then split the collected data into train, validation, and test sets using the 8:1:1 ratio. The dataset statistics are summarized in Table 1, including the number of dialogues, utterances, average length, as well as data sources used. The average length per utterance reaches 17.6, hinting at the informativity and depth of the conversation. It is shown that over 86% of responses used persona (*i.e.*, resp w/ persona) and 76% used both persona and knowledge (*i.e.*, resp w/ p_and_k), which shows that KBP is capable as a benchmark to evaluate the different grounding abilities of models.

## 4 Method

### 4.1 Task Definition

We first provide a general definition of a dialogue system that requires multiple sources and then we instantiate the definition in the context of personalized knowledge-grounded dialogue. For each dialogue, the dialogue context $c = \{u_1, s_1, u_2, s_2, ..., u_t\}$ and different knowledge sources $\boldsymbol{K} = \{K_1, K_2, ..., K_i\}$ are provided, where $K_i = \{k_i^1, k_i^2, ..., k_i^j\}$ indicates the $i_{th}$ source's name of $\boldsymbol{K}$. $k_i^j$ denotes the $j_{th}$ knowledge in natural language from $K_i$. If $K_2$ is reliant on $K_1$, knowledge should be retrieved from $K_2$ based on the selected knowledge in $K_1$. Such reliance should also be embodied in the construction

---

[9]The knowledge related to the same persona forms a document, and different documents form DOCUMENTS source.

of $K_2$, in a way such as $K_2 = \{k_1^1 : \{k_2^1, k_2^2\}, k_1^2 : \{k_2^3, k_2^4\}, ...\}$. The goal of the system is to generate a response $s_t$ conditioned on $c$ and a set of knowledge $\{k_i^j, ..., k_n^m\}$ retrieved from $K$ if required[10]. Specifically in the context of personalized knowledge-grounded dialogue, we regard $\{\text{PERSONA}, \text{DOCUMENTS}\}$ as $\{K_1, K_2\}$ respectively. There is a potential dependency between these two sources, and the goal is to generate a response $s_t$ conditioned on a set of knowledge $\{p_i^j, ...k_m^n\}$, where $p_i^j$ is retrieved from PERSONA and $k_m^n$ is retrieved from DOCUMENTS. The response can also be generated conditioned on a set of knowledge from a single source PERSONA or without any sources.

## 4.2 Supervised *SAFARI*

There are three different steps in our proposed SAFARI framework: ***Planning***, **Retrieval**, and **Assembling** as shown in Figure 1(b). We will introduce them step-by-step under both supervised and unsupervised settings.

***Planning***   The goal of the planning step is to make a series of decisions to decide whether or not the corresponding source of knowledge is required and determine their call order if needed. Since the dependency relationship is previously known, we only need to make sure that a certain knowledge source is called after the sources it depends on. Thus, we formulate this task as sequence-to-sequence generation by directly outputting either required sources in execution order or NULL if the response does not need any knowledge as follows:

$$\mathcal{M} : c \to K_i, K_j, ..., K_n \quad or \quad \text{NULL}, \quad (1)$$

where $\mathcal{M}$ is parameterized by LLMs. We add $K_i, ..., K_n$ and NULL into the vocabulary of LLMs as special tokens. The key idea here is similar to the recent ToolkenGPT (Hao et al., 2023), which regards different tools as special tokens in the vocabulary. Besides that, we add other special tokens to indicate the different parts of the input, *i.e.,* [SOURCE] and [EOS] to indicate the start and end positions of sources. In this way, LLM can model the dependency between different sources and learn when and how to call certain sources.

***Retrieval***   According to the output of the last step, there are two cases in this step: (1) the response does not need any external sources of knowledge,

[10]Unlike most of the previous works, we also consider the response that does not need any knowledge in our setting.

---

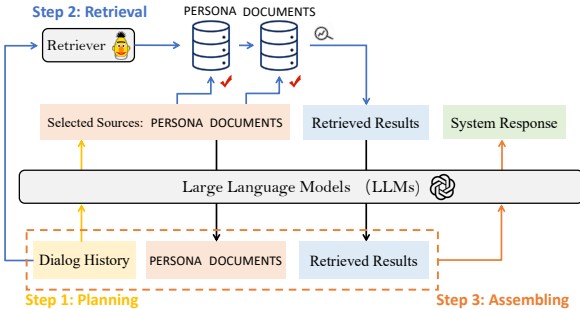

Figure 2: The supervised framework of **SAFARI** for personalized knowledge-grounded dialogues. We use different colors to indicate different steps. The black arrow denotes the flow of data without the the involvement of LLM.

and the agent can skip this step; (2) the response needs multiple sources of knowledge, and the agent strictly follows the output source order to retrieve *top-n* related knowledge $k_i^*$ for the $i_{th}$ knowledge source according to the dialogue context $c$, and if there is a dependency here, it will use preceding retrieved results $k_j^*$ in the planned execution order as a filter. Specifically, assuming the output order is PERSONA, DOCUMENTS in the planning step for a persona-consistent dialogue system, we first retrieve *top-1* result $p^*$ from PERSONA, and then we retrieve $k^*$ from DOCUMENTS according to $c$ and $p^*$. Here the utilization of $p^*$ depends on the systematic design. For example, if there is a designated source $K^*$ for each $p^*$ (a.k.a dependency), we can simply retrieve $k^*$ from $K^*$. And there is another case where all knowledge is stored together and we can concatenate $c$ and $p^*$ as a query in the retriever.

$$\mathcal{R} : K_i, K_j, ..., K_n \to k_i^j, ..., k_n^m \quad (2)$$

***Assembling***   We concatenate all preceding results together with the dialogue context $c$ to generate the response:

$$\mathcal{M} : Inp \to s_t, \quad (3)$$

where $Inp = \{c \quad \text{[SOURCE]}K_i, ..., K_n\text{[EOS]} \text{[MIDDLE]}k_i^j, ..., k_n^m\text{[EOM]}\}$. [MIDDLE] and [EOM] represent the start and end positions of retrieved results. Forming the input in this way has two advantages. Firstly, the name of the sources indicates the type of results retrieved, which provides more signals to the LLMs. Secondly, it allows us to train the language model in a multi-task manner using teacher forcing. The loss is only calculated on tokens related to the planning and the response as shown in Figure 2. We first predict the planning

and then generate the response according to the preceding results when inference.

## 4.3 Unsupervised *SAFARI*

Inspired by the recent progress using LLMs as a controller to plan a call order of different models (Shen et al., 2023), we adopt a similar way here by providing detailed prompts to leverage the LLMs' capability to address the dependency between different knowledge sources. We consider two settings: zero-shot and in-context learning for planning and assembling steps here since the retrieval step is the same as above.

---

There are different knowledge bases storing relevant information:
**K_1: {K_1_DESC}**
**K_2: {K_2_DESC}**
......
There exists a dependency between these knowledge bases.
**{DEPENDENCY_DESC}**
Here is the dialogue between the user and the system: **{DIALOGUE}**
Based on the user's last question, please determine if it requires invoking the corresponding knowledge base. If the invocation is necessary, output the names of the knowledge bases in the order they should be invoked. If no invocation is needed, output **NULL**.

---

Table 2: The zero-shot prompt of unsupervised SAFARI at planning step (translated from Chinese to English).

---

The dialogue is as follows:
**{DIALOGUE}**
The following knowledge is retrieved from different sources of knowledge bases:
**{MIDDLE_RESULTS}**
Please play the role of the system and generate a reply according to the context of the dialogue and given knowledge. Please make sure your reply is consistent with the given knowledge. If the provided knowledge is **NULL**, generate a response solely based on the dialogue context.
**System:**

---

Table 3: The zero-shot prompt of unsupervised SAFARI at assembling step (translated from Chinese to English).

***Planning.*** Instead of directly providing supervision signals, we provide a description for each source of knowledge, accompanied by the corresponding dependency between the sources. The prompts are shown in Table 2.

***Assembling.*** We feed the dialogue content and the retrieved knowledge into the prompt as organized in Table 3, adapting LLMs to generate responses according to dialogue context and retrieved results.

The full prompts of the unsupervised planning step can be found in Appendix D. For few-shot in-context learning, we prepend three corresponding demonstrations from the train set to the zero-shot prompts during evaluation.

## 5 Experiments

### 5.1 Experimental Setups

**Implementation Details.** We mainly choose BELLE-LLAMA-7B-2M (Ji et al., 2023) and ChatGLM-6B (Du et al., 2022; Zeng et al., 2023) as two backbone models for supervised setting since they are two popular open-source Chinese models. And we additionally add ChatGPT (gpt-3.5-turbo-0301)[11] for the unsupervised setting. For training, we set the batch size as 8, train models with 3 epochs and save the checkpoint with the lowest validation loss. For other hyper-parameter settings, we mainly follow the corresponding official code[12]. Due to the computation limit, we conduct training with LoRA (Hu et al., 2021) at one single 3090 GPU, and it cost about 4-6 hours. For the unsupervised setting, we set both the temperature and top p as 0.1 to reduce the randomness of LLMs. Besides that, we use three types of retrievers including both sparse and dense retrieval: BM25 (Robertson and Zaragoza, 2009), DPR (Karpukhin et al., 2020), and RocketQAv2 (Ren et al., 2021). We only retrieve the top-ranked result from each source in the experiments[13].

**Evaluation Metrics.** During the evaluation, we use different metrics at different steps. We use F1 for planning, Recall@1 for retrieval, and BLEU (Papineni et al., 2002), Rouge-L (Lin, 2004). For assembling, Knowledge Consistency (K.C) and Persona Consistency (P.C) are calculated using our finetuned NLI models (Madotto et al., 2019). More details including retrieval models, NLI models, and NLI metrics can be found in Appendix C.

### 5.2 Performance of *Planning*

There are three types of decisions representing different sources required in the next step: NULL, PERSONA, and Both (selecting both PERSONA and DOCUMENTS). Table 4 demonstrates the F1 of planning under different settings. Under supervised settings, despite LLMs achieving high F1 scores at Both, the performance at NULL and Persona is still unsatisfactory, since there are fewer training samples in these two cases. On the other hand, under

---

[11]https://openai.com/blog/chatgpt
[12]https://github.com/THUDM/ChatGLM-6B and https://github.com/LianjiaTech/BELLE
[13]The effects of different choices is analyzed at Section 6.

| Model | NULL | Persona | Both |
|---|---|---|---|
| *Supervised* | | | |
| BELLE-LLAMA-7B-2M | 42.67 (194) | 14.08 (17) | 83.77 (1018) |
| CHATGLM-6B | **47.10** (129) | **31.96** (69) | **86.59** (1031) |
| *Unsupervised* | | | |
| *Zero-shot* | | | |
| BELLE-LLAMA-7B-2M | **28.55** (940) | 8.94 (54) | 32.47 (235) |
| CHATGLM-6B | 25.60 (1225) | 0.0 (0) | 0.43 (4) |
| CHATGPT | 11.45 (116) | **20.67** (233) | **74.88** (880) |
| *In-context* | | | |
| BELLE-LLAMA-7B-2M | 9.22 (36) | 18.21 (1193) | 0.0 (0) |
| CHATGLM-6B | 25.67 (1190) | 1.49 (9) | 4.62 (30) |
| CHATGPT | **27.95** (699) | **23.14** (238) | **41.98** (292) |

Table 4: The F1 of different decisions in **Planning** of different LLMs under supervised/unsupervised settings. We also report the frequency of different decisions in the bracket. There are 181 NULL, 125 PERSONA and 923 PERSONA, and DOCUMENTS in the ground planning.

unsupervised settings, the LLMs are over-confident in their decisions to use NULL, and they misunderstand the dependency between different sources (sometimes deciding to only use DOCUMENTS without PERSONA)[14]. This result reveals the LLMs' low accuracy in expressing uncertainty and fetching unknown knowledge. Furthermore, in-context learning cannot improve this situation, which is similar to the observation in Amayuelas et al. (2023).

## 5.3 Performance of *Retrieval*

With the ground-truth planning labels (except NULL), we examine three types of retrievers, including **BM25**, **RocketQAv2**, and **DPR**, to evaluate the retrieval performance. Table 5 presents the Recall@1 (R@1) of the different retrievers. We found that the DPR and RocketQAv2 can achieve over 80% R@1 when retrieving from PERSONA source while only about 50% from DOCUMENTS and the R@1 at DOCUMENTS† further decreases after removing the dependency. First, the semantics between different knowledge from DOCUMENTS with the dependency are similar to the same underlying persona $p^*$, making them more difficult to be distinguished. In addition, noisy knowledge sentences are introduced since there exists no dependency. Moreover, we observe that DPR performs the best out of these three retrievers in all sources of knowledge while BM25 performs worst[15], revealing the importance of dense retrieval models in this task.

---

[14]We assign the case that LLMs predict DOCUMENTS only as NULL since this case does not exist in KBP.

[15]RocketQAv2 is generally not competitive with DPR because of the pre-trained weights in the RocketQAv2, since it is pre-trained using QA datasets and the length of the question is much shorter than dialogue context.

| Model | Persona | Both | | |
|---|---|---|---|---|
| | | PERSONA | DOCUMENTS | DOCUMENTS† |
| BM25 | 36.80 | 48.97 | 15.05 | 11.37 |
| RocketQAv2 | 80.00 | 92.31 | 50.49 | 35.75 |
| DPR | **83.20** | **93.07** | **51.67** | **39.33** |

Table 5: The performance of **Retrieval** of different types of retrievers. There are 125 examples that only require PERSONA and 923 require both PERSONA and KNOWLEDGE. We also report the Recall@1 of DOCUMENTS without dependency (DOCUMENTS†).

Therefore, we set **DPR** as the retriever in our experiments afterward.

## 5.4 Performance of *Assembling*

Table 6 demonstrates the performance of response generation under both supervised and unsupervised settings. Referring to Table 4, the performance of the planning step largely affects the results in the assembling step, when the retriever is the same. Mostly, better planning leads to better responses in all metrics. The supervised models are much better than unsupervised models since their planning results are much better, while ChatGPT performs best under unsupervised settings due to a similar reason. We found that BELLE achieves higher BLEU1 and Rouge-L, K.C but lower P.C than ChatGLM since the planning gap between them mainly comes from PERSONA. In addition, due to poor retrieval performance at DOCUMENTS (Table 5), the consistency score K.C is also much lower than P.C. With demonstrations in the prompt, we observe generally better performance on most metrics, since LLMs tend to accept the personalized role, rather than generating responses like "*As an AI language model, I do not have persona ....*". Overall, we conclude that the grounding ability of supervised models is much better than unsupervised ones, and ChatGPT performs best under the unsupervised setting.

## 6 Analysis

In this section, we analyze the effects of different components and the choice of the number of retrieved results, based on ChatGLM under the supervised setting. In addition, we conduct human evaluations to verify the quality of automatic evaluations.

## 6.1 Impacts of Different Steps

We investigate the effects of individual steps by providing the model ground-truth labels from each step to generate the response, enabling us to an-

| Model | BLEU1 | Rouge-L | P.C | K.C |
|---|---|---|---|---|
| *Supervised Setting* | | | | |
| BELLE-LLAMA-7B-2M | **30.48** | **34.61** | 75.34 | **46.62** |
| CHATGLM-6B | 23.81 | 26.70 | **76.99** | 42.39 |
| *Unsupervised Setting* | | | | |
| *Zero-shot* | | | | |
| BELLE-LLAMA-7B-2M | 11.84 | 19.24 | 30.59 | 27.34 |
| CHATGLM-6B | 6.18 | 14.50 | 14.73 | 24.73 |
| CHATGPT | 12.06 | 24.44 | **73.47** | **38.00** |
| *In-context* | | | | |
| BELLE-LLAMA-7B-2M | **19.51** | 22.25 | 72.98 | 24.89 |
| CHATGLM-6B | 13.74 | 19.69 | 16.92 | 24.89 |
| CHATGPT | 16.03 | **25.62** | 46.38 | 35.56 |

Table 6: The performance of **Assembling** under supervised/unsupervised settings.

| Model | BLEU1 | RougeL | P.C | K.C |
|---|---|---|---|---|
| CHATGLM-6B | 23.81 | 26.70 | 76.99 | 42.39 |
| + Ground Planning | 24.29 | 27.01 | 86.16 | 57.12 |
| + Ground Retrieval | **25.86** | 29.15 | 79.52 | 53.95 |
| + Ground P & R | 25.71 | **29.43** | **90.56** | **72.99** |
| - Dependency | 23.32 | 25.53 | 75.67 | 38.49 |
| - Documents | 23.06 | 25.34 | 75.91 | 36.53 |
| - Planning* | 23.51 | 25.98 | 72.90 | 24.89 |
| - Planning** | 23.69 | 26.81 | 71.60 | 34.91 |

Table 7: Ablation study on the impact of different steps and modules in SAFARI.

| Number | Assembling | | | |
|---|---|---|---|---|
| | BLEU1 | RougeL | P.C | K.C |
| 1 | **23.81** | **26.70** | **76.99** | **42.39** |
| 2 | 22.70 | 25.57 | 71.03 | 29.45 |
| 3 | 20.69 | 24.05 | 69.73 | 27.91 |

Table 8: The performance of **Assembling** of different number of retrieved results.

alyze and understand the specific effects of each step in a clear and systematic way. Table 7 presents the results. First, we note that the inclusion of ground-truth planning labels or knowledge casts a positive impact on performance. Planning primarily enhances P.C and K.C, while grounding knowledge contributes to BLEU1 and Rouge-L scores. The best results are obtained when both signals are combined. Secondly, we also conduct an ablation study by removing some modules: 1) removing dependency information (-Dependency); 2) removing DOCUMENTS and only using PERSONA (-Documents); and 3) removing the planning step by always selecting PERSONA (-Planning*) or always selecting PERSONA and DOCUMENTS (-Planning**) for each turn. It can be found at the bottom of Table 7 that all metrics are dropped differently after removing different components except for Rouge-L when always selecting two knowledge sources. To conclude, SAFARI can effectively incorporate multiple sources (compared with -Documents) and further address dependency issues (compared with -Dependency). Moreover, SAFARI demonstrates its versatility by effectively handling multiple sources and efficiently selecting relevant ground knowledge. Notably, SAFARI outperforms existing methods that indiscriminately utilize all available sources (compared with -Planning**).

## 6.2 Different Numbers of Retrieved Results

The number of retrieved results plays a key role in the response generation. There is a trade-off between accuracy and recall, while a small number of retrieved results may not cover enough semantics but a large number may introduce additional noises. Table 8 presents the results of the different numbers of retrieved results. We observe that the

performance of response generation decreases with the number, which indicates that noisy knowledge will harm the quality of the generated responses.

## 6.3 Human Evaluation

Human evaluation is conducted to evaluate the quality of generated response in terms of three metrics: coherence score (**Coh.**), persona consistency score (**Per.Cs**), and knowledge consistency score (**Know.Cs**). We randomly sample 100 responses with grounding information for each model and ask three annotators to indicate its coherence score (1-5) and whether the response is consistent with the given persona (1/0), and knowledge (1/0). Table 9 shows the result. We observe that supervised methods achieve higher performance than unsupervised ones, which corroborates the findings of the automatic evaluation results presented in Table 6. Besides that, we found BELLE achieves the highest performance across all metrics and outperforms ChatGLM since the effects of planning are not considered during human evaluation. Moreover, we also found that in-context learning brings a lower rejection rate and more human-like responses. Specifically, the rejection rate of BELLE under the setting of zero-shot learning is about 32%, while the number is reduced to 12% under in-context learning[16].

## 7 Conclusion

In this paper, we propose a novel framework SAFARI to incorporate multiple sources of knowledge

---

[16]More analysis can be found in Appendix E

| Model | Coh. | Per.Cs (%) | Know.Cs (%) |
|---|---|---|---|
| **Supervised Setting** | | | |
| BELLE-LLAMA-7B-2M | 4.38 | 72.0 | 63.8 |
| CHATGLM-6B | 4.06 | 68.0 | 59.1 |
| **Unsupervised Setting** | | | |
| *Zero-shot* | | | |
| BELLE-LLAMA-7B-2M | 2.84 | 24.7 | 19.5 |
| CHATGLM-6B | 2.58 | 17.0 | 14.8 |
| CHATGPT | 4.00 | 63.4 | 33.3 |
| *In-context* | | | |
| BELLE-LLAMA-7B-2M | 3.36 | 40.0 | 21.7 |
| CHATGLM-6B | 2.88 | 32.0 | 28.9 |
| CHATGPT | 4.03 | 54.0 | 48.8 |

Table 9: The results of human evaluation. The inter-agreement is about 86%.

bases and further address the dependency issue between them. Unlike previous works, SAFARI can be extended to multiple sources easily and it can handle cases that do not require any sources or require some instead of all sources between them. We build the first personalized knowledge-grounded dialogue (KBP) dataset, and experimental results prove the effectiveness and robustness of SAFARI.

## Limitations

In this paper, we propose the SAFARI framework to incorporate multiple sources and address the dependency issue between them by leveraging the exceptional capability of LLMs. However, we acknowledge two major limitations of the paper:

**SAFARI** There is an error propagation issue when decoupling the source selection from the response generation. The cascaded design may propagate the error in the intermediate step. Once the planning sources are wrong, the retriever can not retrieve correct results from the knowledge bases.

**KBP** Although the constructed dataset has already considered three different cases in which responses require 0, 1, and 2 knowledge sources, there are other useful sources in the knowledge-grounded dialogue system such as the users' memory source or other sources in different applications. Besides that, the world knowledge may become outdated as time goes by.

## Ethics Statement

In this paper, there are two ethical issues about the LLMs and dataset respectively.

**Usages of LLMs** We strictly follow the license and policy of released LLMs, and we do not guarantee the content generated content by LLMs is

safe and harmless. We note that LLMs may inherit hallucination issues as shown in the planning analysis, and it will plan not to use corresponding sources due to poor performance to express uncertainty. The calls of the OpenAI API in this paper were carried out by Dr. Yang Deng, one of the corresponding authors from the National University of Singapore.

**Human Annotation** The human inspection and annotation were conducted by a reputable data annotation company, and the annotators are compensated fairly based on the market price without revealing any personal information. Besides that, our dataset may contain biased opinions due to the subjectivity of manual annotation.

## Acknowledgement

We would like to express our heartfelt gratitude to all anonymous reviewers for their insightful comments and suggestions. The work described here was partially supported by grants from the RGC General Research Funding Scheme (GRF) 14222922 (CUHK 2151185).

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

# A  Dialogues Requiring Knowledge Dependency

Table 10 shows another example of knowledge dependency. In this example, without considering the persona-knowledge reliance, the entity (proverb 瓦合), which is irrelevant to the band 草东没有派对, is retrieved. Although the retrieved knowledge is unrelated to the ground persona, ChatGPT still tries to combine these two disconnected pieces of information and generate an inconsistent response (**Response w/o. Dependency**).

---

**Ground Persona:** "I really like 草东没有派对 *(No Party for Cao Dong)*"
**Retrieved Knowledge w/o. dependency:** "瓦合 *(The Clod)* is a Chinese proverb that metaphorically represents a motley crowd."
**Ground Knowledge w. dependency:** "瓦合 is a new album from the band 草东没有派对"

======

......
**User:** Do you want to listen to 瓦合 *(The Clod)*?
**Response w/o. Dependency:** I absolutely love 草东没有派对! 瓦合 is a proverb that describes a motley crowd, representing a group of people without organization or coordination. So regarding 瓦合, I can say that it represents a group of individuals lacking organization or coordination. Do you have any other questions I can assist you with?
**Response w. Dependency:** You bet! I absolutely love 草东没有派对! 瓦合 is their newly released album, and I'm really looking forward to listening to the songs on it. Would you like to share with me which song you enjoy the most in 瓦合.
......

---

Table 10: An example dialogue with ChatGPT showing persona-knowledge dependency (translated from Chinese to English)

# B  Data Preparation

**Seed Preparation** In the original persona dataset, DuLeMon, most of the persona does not contain much knowledge. For example, about 30% to 40% of persona is about the name such as *"my name is XXX"*. After clustering, we removed topics that contain less than 20 personas. There are 5 topics left: *career, book, music, movie, personality*. Then we add some basic persona such as *gender, age, locality, nation, sport, and others*.

**Persona and Knowledge Matching** For each persona description after segmentation, we first remove stop-words[17] and frequently used words.

---

[17] https://github.com/goto456/stopwords/blob/master/cn_stopwords.txt

| Attributes | Templates |
|---|---|
| nick_name, birth_place, ... | {head entity}'s {attribute} is {tail entity} |
| rewards, members, ... | The {attribute} of {head entity} contains {tail entity} |

Table 11: Two major templates to convert triples into natural language descriptions (translated from Chinese to English).

And then we filter duplicated words to reduce unnecessary computation. For the left words, we map them to the corresponding knowledge base one by one. We discarded some useless attributes such as *penmanship* and *pronunciation*, and then translate the matched triples to natural language sentences according to pre-defined templates as shown in Table 11. If we do not find any matched triples, we just discard the persona. At last, we manually checked each persona description to make sure (1) there is no repetitive knowledge statement, and (2) each persona contains at least 5 knowledge statements [18].

# C  Implementation Details

We first illustrate the details of finetuning and then introduce our definition of NLI metrics: P.C and K.C.

**Retrieval Models.** We finetune RocketQAv2 and DPR using our own KBP dataset by regarding *(context, used_persona / document)* as the positive and *(context, unrelated_persona / document)* as the negative. We set epochs as 5 and max sequence length as 512, and mainly follow the scripts: https://github.com/PaddlePaddle/RocketQA and https://github.com/Alibaba-NLP/Multi-CPR/tree/main/retrieval respectively for other parameters. For RocketQAv2, we load the weights of pre-trained model zh_dureader_de_v2 as introduced in the official homepage, which is trained on the largest Chinese QA dataset, and we use 12-layer bert-base-chinese with 110M parameters as backbone model for DPR.

**NLI Models.** Following previous work (Kim et al., 2020; Cao et al., 2022), we finetune an NLI model (Welleck et al., 2019) using our own dataset by regarding *(ground_persona / document, response)* as the positive and randomly sampled *(unrelated_persona / document, response)* as the negative. We also use bert-base-chinese as the backbone model. We concatenate and encode the ground persona/document $k$ and response $r$ in the form of $[CLS]k[SEP]r[SEP]$, and we train the

---

[18] We will manually add it if there are less than 5 knowledge statements.

$$\mathbf{C}(r, g) = \begin{cases} \mathbf{NLI}(r, g), & \text{if there is grounding } g \text{ for } r, \text{ and planning also uses } g'. \\ 0, & \text{if there is grounding } g \text{ for } r, \text{ and planning } \textbf{does not} \text{ use } g'. \\ 0, & \text{if there is } \textbf{no} \text{ grounding } g \text{ for } r, \text{ and planning uses } g'. \\ 1, & \text{if there is } \textbf{no} \text{ grounding } g \text{ for } r, \text{ and planning } \textbf{does not} \text{ use } g'. \end{cases} \quad (4)$$

model to predict whether responses are consistent with corresponding personas or documents. The batch size for fine-tuning is 8. The maximal training epoch is 5, and the max sequence length of the encoder is 512. In the experiments, we use the AdamW optimizer with a learning rate of 2e-5 and an epsilon of 1e-6. We evaluate the NLI model on the KBP test set every 500 iterations during training, and we save the checkpoint with the highest performance on the KBP test set. The fine-tuned model achieves $> 95\%$ accuracy for both persona ($95.94\%$) and knowledge ($96.95\%$) on the KBP test dataset. We then calculate the persona consistency (P.C) and knowledge consistency (K.C) according to the output of the NLI model. Compared with many previous works (Madotto et al., 2019; Cao et al., 2022) that calculate the NLI score only when ground-truth knowledge includes personas or documents, our framework is more sophisticated and introduces the planning step, which considers the situation where responses do not require any ground knowledge. Thus, if we only calculate the consistency score in the occasion where there exists ground truth personas/documents, it is unfair and inaccurate for our framework, since wrong ground knowledge could also hurt the quality of responses and the system is not penalized when it gives wrong planning (*e.g.* always calling external sources). We design a new calibrated metric for dialogue consistency as described in Eq 4,5,6,7:

$$\mathbf{NLI}(r, g) = \begin{cases} 1, & \text{if } r \text{ entails } g \\ 0, & \text{if } r \text{ does not entail } g \end{cases} \quad (5)$$

$$\mathbf{P.C} = \frac{\sum_i^m C(r_i, p_i)}{m} \quad (6)$$

$$\mathbf{K.C} = \frac{\sum_i^m C(r_i, k_i)}{m} \quad (7)$$

There are four cases during experiments as shown in Eq 4: 1). there exists ground-truth grounding $g$ for the response $r$ in the test set and the planning also decides to retrieve ground knowledge $g'$ from corresponding sources (either PERSONA or DOCUMENTS); 2). there is $g$ for $r$, but the planning

decides not to use $g'$; 3). there is no grounding $g$ for $r$, while the planning decides to use $g'$; 4). there exists no $g$, and the planning step decides not to use $g'$ either. We only calculate the NLI score of $(g, r)$ using finetuned NLI model in the first case of Eq 4. With this definition, we can get the calibrated score **P.C** and **K.C** for the KBP test set.

## D   Prompts Templates under Unsupervised Setting

Table 12 demonstrates the zero-shot prompt of the SAFARI planning step on the KBP dataset, and the zero-shot prompt for the assembling step is the same as Table 3.

There are two knowledge bases storing relevant information:
**PERSONA**: This knowledge base stores information related to system personas, such as gender, age, place of origin, hobbies, personality traits, and other relevant data.
**DOCUMENTS**: This knowledge base stores domain-specific knowledge related to system personas, such as the domain knowledge about the place of origin.
There exists a dependency between these knowledge bases. The invocation of DOCUMENTS relies on the results from PERSONA. Please ensure the correct order of invoking them.
Here is the dialogue between the user and the system: **{dialogue_history}**
Based on the user's last question, please determine if it requires invoking the corresponding knowledge base. If the invocation is necessary, output the names of the knowledge bases in the order they should be invoked. If no invocation is needed, output NULL.

Table 12: The prompts of unsupervised SAFARI on KBP dataset  (translated from Chinese to English)

## E   Human Evaluation

We find that human is more likely to find persona-inconsistent cases. There are some responses that have intra-sentence contradictions (Zheng et al., 2022), for example, "*My zodiac signs are Aries and Taurus*". In addition, there are other responses related to the persona description that are inconsistent. Both these types of responses are easy to identify by humans but hard for the NLI model to detect, resulting in lower Per.Cs during human evaluation.