# OpenReview forum: "Large Language Models as Source Planner for Personalized Knowledge-grounded Dialogues"
_EMNLP/2023/Conference — EMNLP 2023 Findings_

### Official Review · Reviewer_C1P9 · 2023-08-04

**Soundness:** 4

**Excitement:**

3: Ambivalent: It has merits (e.g., it reports state-of-the-art results, the idea is nice), but there are key weaknesses (e.g., it describes incremental work), and it can significantly benefit from another round of revision. However, I won't object to accepting it if my co-reviewers champion it.

**Paper Topic And Main Contributions:**

The paper mainly addresses the issue of incorporating multiple knowledge sources and their dependency when generating knowledge-grounded dialogues. The author proposed a framework named SAFARI, leveraging LLMs to first generate the person and relevant documents and then incorporate the retrieved documents and persona into response generation. To study the effectiveness, the author collected a new personalized dialogue dataset called KBP. The experimental results show that the proposed framework is effective in incorporating different knowledge sources.

**Questions For The Authors:**

1) How do you control the quality of the dataset? What measures do you take?
2) How do you measure the consistency between different annotators during human evaluation?

**Reasons To Accept:**

1) The paper reads well and is easy to follow.
2) The collected dataset might be a useful resource for the community to do research on personalized dialogue.
3) The paper proposed a novel framework to address the multiple knowledge sources incorporation issue.


**Reasons To Reject:**

1) The size of the dataset is somewhat small with only 2477 dialogues. It would be more useful if the author can increase the size of the dataset.
2) More experiments on other benchmark datasets would be nice to show the generalization ability of the proposed framework.

**Reproducibility:**

4: Could mostly reproduce the results, but there may be some variation because of sample variance or minor variations in their interpretation of the protocol or method.

**Reviewer Confidence:**

4: Quite sure. I tried to check the important points carefully. It's unlikely, though conceivable, that I missed something that should affect my ratings.

---

> ### Author Rebuttal · Authors · 2023-08-28
>
> Thank you for your positive reviews and we will address your concerns one by one as shown in the following:
>
> **Reasons to Reject:**
>
> 1. Thank you for your suggestions. We think the size of KBP is relatively large enough (also supported by reviewer qWAp) for current LLMs to learn the planning and grounding capabilities, but we would like to increase the number of sources of knowledge in future work.
>
> 2. As the **first work** to explore the planning capability of LLMs in the context of multi-source knowledge-grounded dialogue generation (e.g., decide whether or not require knowledge, which source to call, and when to call), there are **no other suitable datasets** for us to conduct experiments (e.g., the dependency between different sources of knowledge; and contains responses that do not use knowledge and use multiple sources of knowledge simultaneously), Thus we mainly conduct it on our own datasets. We also want to note that the persona and documents in the test datasets **are different** from the training datasets of KBP, which proves the generalization ability of SAFARI.
>
> **Questions for the Authors:**
>
> 1. We adopt several methods to control the quality of the dataset: 1) training and pilot annotation (lines 238 to 255); 2) batch collection (lines 256 to 263); 3) We also conduct automatic evaluation and human evaluation to control the quality as introduced in footnotes 7 and 8.
>
> 2. As shown in the caption of Table 9, we use inter-agreement to measure the consistency between human annotators.

---

### Official Review · Reviewer_WKZY · 2023-08-05

**Soundness:** 3

**Excitement:**

2: Mediocre: This paper makes marginal contributions (vs non-contemporaneous work), so I would rather not see it in the conference.

**Missing References:**

OpenDialKG: Explainable Conversational Reasoning with Attention-based Walks over Knowledge Graphs - Moon et al.

**Paper Topic And Main Contributions:**

In this paper, the authors introduce a framework that aims to improve dialogue systems by allowing them to incorporate multiple knowledge sources into their responses. Specifically, they explore how LLMs can facilitate the connection between various knowledge sources in the context of personalized knowledge-based dialogue systems. To support this research, the authors created a dataset called KBP which is curated for personalized knowledge-based dialogues. What sets this dataset apart is that the responses are influenced by multiple knowledge sources, creating more engaging and informative conversations that align with the user's knowledge and persona.

**Questions For The Authors:**

1. In Tables 6 and 9, it is observed that ChatGPT exhibits lower P.C/K.C/persona consistency scores after the addition of in-context learning examples. Could you provide some insights into why this might be the case? What factors do you believe contributed to this decrease in performance?

2. To further substantiate the necessity of your planning and assembling strategy, have you considered conducting an experiment where ChatGPT is used without these stages? Could you share any findings or observations from such an experiment, if it was conducted?

**Reasons To Accept:**

1. The paper introduces the KBP dataset. It could be a unique resource specifically for personalized knowledge-grounded dialogues. The dataset's focus on responses conditioned on multiple knowledge sources makes it a valuable asset for future research in this area. This could potentially lead to the development of more sophisticated and user-centric conversational AI applications.

2. The authors conduct comprehensive and meticulous experiments, including detailed ablation studies. The reported performance is impressive.

**Reasons To Reject:**

1. While the paper's focus on integrating multiple knowledge sources is commendable, it falls short of demonstrating the practical application of this approach. Specifically, it lacks empirical results that showcase the combined use of all knowledge sources into a single one and then use established frameworks such as the RAG for retrieving knowledge from a single source. This omission limits the depth of the study and the potential for comparative analysis with existing methodologies.

2. The absence of human performance benchmarks in Tables 4 and 6 is a significant oversight. Including human performance metrics would provide a more comprehensive understanding of the system's effectiveness and its comparative standing against human-level performance.

3. The paper could have benefited from an exploration of how LLMs such as ChatGPT perform without the planning and resembling stages, especially in the context of Table 7. Given that LLMs inherently incorporate various forms of knowledge.

**Reproducibility:**

4: Could mostly reproduce the results, but there may be some variation because of sample variance or minor variations in their interpretation of the protocol or method.

**Reviewer Confidence:**

3: Pretty sure, but there's a chance I missed something. Although I have a good feel for this area in general, I did not carefully check the paper's details, e.g., the math, experimental design, or novelty.

**Typos Grammar Style And Presentation Improvements:**

When reporting inter-agreement, it is recommended to instead report the kappa value in Table 9.

---

> ### Author Rebuttal · Authors · 2023-08-28
>
> We greatly appreciate your valuable suggestions and feedback. We carefully address your concerns one by one as follows:
>
> **Reasons to Reject:**
>
> 1. We appreciate the comment and we also acknowledge the organization of different sources of knowledge is valuable as discussed below. However, we would like to clarify that this work focuses more on **the planning capability of LLMs to decide whether or not require knowledge, which source to call, and when to call.** For your problem here for our paper, there are four different ways to organize PERSONA and DOCUMENTS: $a, b, c, d$ as defined below. We provide the comparison results of $a$ and $b$ in Table 7 in the paper and we provide additional results of $c$ and $d$ as you proposed.
>    * $a$: There is a dependency between PERSONA and DOCUMENTS as illustrated in the paper. Suppose N is the size of PERSONA and there are M documents for each persona description in PERSONA (a.k.a., the size of DOCUMENTS is NM), then the time complexity is $O(N+M)$ since we will filter other irrelevant DOCUMENTS in the second step. The space complexity is $O(N+NM)$
>
>    * $b$: There is no dependency between PERSONA and DOCUMENTS as illustrated in the ablation study (Table 7). The time complexity is $O(N+NM)$ and space complexity are $O(N+NM)$
>
>    * $c$: Simply combine PERSONA and DOCUMENTS into one new source. In this way, the size of the new source becomes **the sum of** the size of PERSONA (N) and DOCUMENTS (NM). The time complexity is $O(2(N+NM))$ (2 here refers to the number of retrieved times as explained below) and space complexity is $O(N+NM)$.
>
>    * $d$: Concatenating the individual evidence of PERSONA and DOCUMENTS if they are related. In this way, the size of the new source becomes **the product of** the size of PERSONA and the number of documents for each persona description (M). The time complexity is $O(NM)$ and the space complexity is $O(2NM)$ (here we need to save original N persona NM times, thus 2 NM). The situation becomes worse once the number of sources keeps increasing.
>
>    We use DPR as an encoder and retrieve 2 items in $c$ and 1 item in $d$ for a fair comparison. In $c$, there is a chance that both of the returned 2 items are from PERSONA or DOCUMENTS, we use whether or not the returned results contain the ground persona or knowledge to calculate the metric (**thus this is not exactly recall@1**). The retrieval performance can be found in Table 2. For analysis $a$ and $b$, please refer the Table 7 in the paper. On the one hand, due to introduced noises in $c$, we can find that the performance of PERSONA drops a lot ($83.20 \rightarrow 54.40$ and $93.07 \rightarrow 38.67$), and the improvement of DOCUMENTS is caused by the increased number of returned results since both two returned items are from DOCUMENTS. On the other hand, for setting $d$, we can find a slight improvement (less than 1\%) in PERSONA and DOCUMENTS in Both situations while a relatively bigger drop (more than 2\%) in the Persona situation. Before we present the performance of assembling, we emphasize that **low efficiency and scalability make the $d$ infeasible** when the number of items in different sources of knowledge increases [5] and the number of sources increases. In addition, it is unnecessary to introduce more information if the response only requires one source of knowledge or part of the information in the concatenation. The time complexity and space complexity can be found in the definition.
>
>
>    Table 3 below shows the assembling results of the above four settings. It can be found there is still a gap between $a$ and $d$, especially at the P.C and K.C metric. We attribute P.C. drop to the low performance at Persona during retrieval and K.C. drop to the use of additional noisy knowledge to generate responses that do not require originally.
>
>
>    Lastly, we want to emphasize our framework SAFARI targets the planning of usages of different sources of knowledge including the response that does not require any source of knowledge. It is not related to how these sources of knowledge are organized, we chose settings $a$ and $b$ mainly based on the novel characteristic of our proposed dataset KBP and their low time complexity and space complexity. Furthermore, RAG, as same as most previous knowledge-grounded methods, makes a strong assumption that all responses require the involvement of external knowledge. However, **it is not held in practice**. For example, there are many other dialogue papers that target non-knowledge-grounded tasks. We target more practical situations in **which the responses do not require, or require one source and even multiple sources of knowledge** by investigating the planning and grounding capability of LLMs. In this way, if we merge multiple sources into single one source, **our framework is still useful to plan (e.g., use knowledge or not), retrieve, and assemble**. Our experiments also center around this motivation and we provide extensive analysis and results to support our claim in the paper.
>
> Table 2: The performance of Retrieval on different settings, where we copy results of $a$ and $b$ from Table 4 in the paper.
> | Settings   | Persona | Both-P | Both-D |
> |------------|---------|--------|--------|
> | $a$ (ours) | **83.20**   | 93.07  | 51.67  |
> | $b$ (ours) | 83.20   | 93.07  | 39.33  |
> | $c$        | 54.40   | 38.67  | 65.33  |
> | $d$        | 80.80   | **93.71**  | **52.11**  |
>
> Table 3: Assembling results of different ways to organize the sources of knowledge in SAFARI.
> | Model      | BLEU1 | RougeL | P.C   | K.C   |
> |------------|-------|--------|-------|-------|
> | $a$ (ours) | **23.81** | **26.70**  | **76.99** | **42.39** |
> | $b$ (ours) | 23.32 | 25.53  | 75.67 | 38.49 |
> | $c$        | 23.16 | 26.44  | 72.02 | 38.56 |
> | $d$        | 23.54 | 26.55  | 74.28 | 39.46 |
>
> 2. We argue that the human performance is self-contained. Since our dataset is collected by human annotators, human performance is the best result (or near best) that can be achieved in each metric in both Tables 4 and 6 in the paper.
>
> 3. First of all, we want to point out that ChatGLM is also a good LLM, especially in Chinese datasets [3,4]. We chose it to conduct our experiments since it is open-source (Thus we can fine-tune it!) and also inherently incorporates various knowledge in Chinese. Secondly, we provide ChatGPT's performance as you suggested below. We note that the general trends are similar to Table 7 in the paper, such as the metrics of -Planning$^{**}$ are mostly better than -Planning$^*$ except P.C. In addition, the P.C. gap between them is also small same as Table 7. **We emphasize that the first row here can not be regarded as the first row in Table 7 since the model in Table 7 is finetuned.** Furthermore, we want to explain more about the highest performance at P.C. There are two reasons: **1) without the training, the planning and grounding capability is not improved for ChatGPT, thus simple planning strategy may get better results than unsupervised zero-shot ChatGPT** (none of the F1 scores of ChatGPT at planning in Table 4 exceeded 75 without training); 2) There are 1229 samples in the test dataset of KBP of which 181 do not require knowledge, 125 require PERSONA and the rest 923 require both sources of knowledge. In this way, the test setting **is more favorable** for simply using all sources or one source here as they are all majority in the test set (> 75\%). Based on these observations, we further validate the claim in the paper (lines 451 to 460: unsupervised LLM can not perform well at planning) and the effectiveness of supervised SAFARI which brings higher planning and grounding capability.
>
> Table 4: The ablation studies of ChatGPT under zero-shot setting
> | Model                   | BLEU1 | RougeL | P.C     | K.C   |
> |-------------------------|-------|--------|---------|-------|
> | ChatGPT (zero-shot)     | **12.06** | **24.44**  | 73.47   | **38.00** |
> |  - Planning$^*$    | 8.17  | 20.36  | 78.19   | 24.89 |
> |  - Planning$^{**}$ | 9.01  | 22.57  | **78.60** | 36.69 |
>
>
> **Questions for the Authors:**
>
> 1. As explained in lines 486 to 490, the performance of the planning step largely affects the results of assembling. From Table 4, it has been observed that the inclusion of in-context examples does not lead to an improvement in the planning performance of language models, and in some cases, it can even result in a decrease in performance. A similar phenomenon is also reported in previous work [6] as explained in lines 451 to 460. Due to bad planning, the ChatGPT exhibits lower scores.
>
>
> 2. Please refer to the above experimental results.
>
> **Missing References and others:**
>
> We will add it to the revised version.
>
> [1] Long time no see open-domain conversation with long-term persona memory. ACL 2022
>
> [2] Prompted LLMs as Chatbot Modules for Long Open-domain Conversation. ACL 2023
>
> [3] C-Eval: A Multi-Level Multi-Discipline Chinese Evaluation Suite for Foundation Models
>
> [4] SuperCLUE: A Comprehensive Chinese Large Language Model Benchmark
>
> [5] There Are a Thousand Hamlets in a Thousand People’s Eyes: Enhancing Knowledge-grounded Dialogue with Personal Memory. ACL 2022
>
> [6] Knowledge of knowledge: Exploring known-unknowns uncertainty with large language models. 2023.5.23

---

### Official Review · Reviewer_EWfx · 2023-08-09

**Soundness:** 3

**Excitement:**

3: Ambivalent: It has merits (e.g., it reports state-of-the-art results, the idea is nice), but there are key weaknesses (e.g., it describes incremental work), and it can significantly benefit from another round of revision. However, I won't object to accepting it if my co-reviewers champion it.

**Paper Topic And Main Contributions:**

This paper focuses on generating a Chinese dialogue dataset, KBP, where responses depend on the persona alone or both the persona and knowledge. Alongside the dataset, it introduces a supervised and unsupervised framework for large language models, SAFARI, that guides the planning, retrieval, and assembly of knowledge and persona when generating a response.

**Questions For The Authors:**

* In Table 1, the resp w/ persona"is 86.1%. What does the other 13.9% represent? Is it NULL or the Knowledge-only case?
* For in-context learning, are the same examples shown to the LLMs? And how many examples are used?
* In Table 7, is CHATGLM-6B (the first row) trained in a supervised setting? Shouldn't the values reported be the same as in Table 6 (the second row)?
* Looking at Table 8, are there responses that deal with more than two documents with persona? If there are no cases using more than two documents in the responses, it makes sense that having more retrieved documents would lead to low performance.


**Reasons To Accept:**

This paper has developed a new persona-knowledge-based Chinese dialogue dataset, KBP. It demonstrates the performance of each step in the proposed SAFARI framework: planning, retrieval, and assembling, and it highlights the effectiveness of these steps.


**Reasons To Reject:**

This paper proposes a new dataset because there are no sufficient datasets available that can capture the knowledge dependency when generating responses. However, the proposed dataset seems quite similar to the FoCus dataset [1]. When creating the FoCus dataset, both persona and knowledge are also combined in responses. The purpose of the FoCus dataset is slightly more oriented towards conveying information rather than engaging in chit-chat. This might differ from the proposed dataset.

The examples in Figure 1 (a) and Figure 10 appear to be general failure cases that could possibly happen, but it seems they are not exact examples from the FoCus dataset [1], which weakens your reason for creating a new dataset. The proposed framework, SAFARI, could potentially be applied to the FoCus dataset. To strengthen your work, supporting it with an empirical analysis or providing failure examples of why the FoCus dataset or another dataset is not sufficient for the persona-knowledge-based dialogue generation task will improve your paper.
(If the examples are directly from a dataset, please let me know)

**Reproducibility:**

3: Could reproduce the results with some difficulty. The settings of parameters are underspecified or subjectively determined; the training/evaluation data are not widely available.

**Reviewer Confidence:**

3: Pretty sure, but there's a chance I missed something. Although I have a good feel for this area in general, I did not carefully check the paper's details, e.g., the math, experimental design, or novelty.

**Typos Grammar Style And Presentation Improvements:**

* In the Evaluation Metric in Section 5.1, "P.C" and "K.C" should be spelled out since they are mentioned for the first time in the paper.

---

> ### Author Rebuttal · Authors · 2023-08-28
>
> We greatly appreciate your valuable comments and suggestions. Regarding all of the concerns you raised:
>
> **Reasons to Reject:**
>
> 1. We noticed and also cited the related work FoCus as you mentioned. To be clear, there are **three unique differences** between KBP and FoCus datasets: **1) The persona and knowledge in FoCus are more like two independent sources of knowledge, while the knowledge is used to convey information. However, the persona and knowledge in KBP are not independent.** To draw a comparison, the FoCus dataset exhibits a parallel relationship between persona and knowledge, where both sources coexist independently. Conversely, in the KBP dataset, a serial relationship emerges, where the persona chosen significantly influences the knowledge presented, and persona-inconsistent responses can arise even without direct reliance on corresponding knowledge [1]; **2) All responses stored in the FoCus dataset require the involvement of knowledge, but there are three different situations in KBP**: responses without requiring any knowledge; response requires one source of knowledge (PERSONA) and response requires two sources of knowledge (PERSONA and DOCUMENTS); 3) The setting is different. In the Focus dataset, the dialogue system is required to choose one knowledge, and some of the persona or not (multiclass classification for knowledge selection and multi-label classification for persona selection in their original paper). However, as we discussed above, KBP is more focused on the planning of different sources, and also once requiring multiple sources of knowledge, it requires the dialogue system to first choose a persona and then related knowledge.
>
> 2. These examples are not from the FoCus dataset since the FoCus dataset does not consider this situation. They are used to illustrate our motivations to propose new settings and found during initial experiments by us and also previous works [1]. We first construct a new dataset to investigate this problem (lines 55 to 77). As we explained above, despite our framework can be applied to the FoCus dataset, the FoCus dataset **can not be used** to investigate the planning capability of LLMs (e.g., decide whether or not requires knowledge, which source to call, and when to call) as it simply always uses knowledge. We compare our method with this kind of strategy in Table 7.
>
> **Questions for The Authors:**
>
> * Yes, the rest 13.9\% is NULL since there are only three situations in KBP (as illustrated in the caption of Figure 1).
> * Yes, they are the same, and we use three examples since there are three situations (lines 405 to 408).
> * Yes, you are right. We carefully checked all files and experimental logs and we believe this is a typo error due to the tight timeline and extensive analysis. The number in Table 7 and Table 8 is correct and we will revise Table 6 accordingly.
> * Yes, there are only 6 examples in our dataset using more than two documents.
>
> **Typos**
>
> Thank you for your suggestions, we will revise it accordingly and hope that our responses can alleviate your concerns.
>
> [1] Like hiking? you probably enjoy nature: Persona grounded dialog with commonsense expansions. EMNLP 2020

---

### Official Review · Reviewer_qWAp · 2023-08-10

**Soundness:** 3

**Excitement:**

3: Ambivalent: It has merits (e.g., it reports state-of-the-art results, the idea is nice), but there are key weaknesses (e.g., it describes incremental work), and it can significantly benefit from another round of revision. However, I won't object to accepting it if my co-reviewers champion it.

**Paper Topic And Main Contributions:**

The paper focuses on the problem of multi-source knowledge grounded dialogue generation. To address this problem, the authors propose a personalized knowledge-grounded dialogue dataset called Knowledge Behind Persona (KBP) and a three-step framework called Source Planner for Personalized Knowledge-Grounded Dialogue (SAFARI), based on a large language model. The experimental results on KBP demonstrate that the SAFARI framework can generate persona-consistent and knowledge-enhanced responses.

**Questions For The Authors:**

A: Could you please clarify the meaning of lines 302-307 in Section 4.1? How does this particular organization of information influence the follow-up method?

B: The framework shown in Figure 1(b) incorporates multi-source knowledge, including persona config, Wikipedia, and memory. However, in the experiments, only the persona config (persona description and reliant knowledge) was utilized. Have you evaluated the framework using real-world applications, such as incorporating other general knowledge sources?

C: Since the persona config is retrieved, how do you ensure persona consistency? Could it be possible to set the persona config (description and reliant knowledge) as fixed to achieve consistent personalized dialogues?

D: Have you evaluated other knowledge-grounded dialogue generation models using the KBP dataset as baselines for comparison?

**Reasons To Accept:**

The paper introduces the KBP dataset, which may be crucial for achieving persona-consistent knowledge-grounded dialogue generation.

The SAFARI framework, consisting of three steps, is proposed and successfully demonstrates its effectiveness in generating persona-consistent responses.

**Reasons To Reject:**

Although the authors declare that the work focuses on multi-source knowledge-grounded dialogue generation, it appears that the knowledge presented in this paper is limited to persona config, including the description and reliant knowledge sentence.

The knowledge within the KBP dataset is limited to persona-related information, making it suitable only for personalized dialogues rather than general knowledge-grounded dialogues like those based on Wikipedia, such as WoW.

Although the data size in KBP is large, the actual knowledge it provides is limited, which may restrict the generalization ability of models trained on this dataset.

The paper only compares the proposed framework with large language model-based approaches, lacking baselines from previous works on knowledge-grounded dialogue.

**Reproducibility:**

4: Could mostly reproduce the results, but there may be some variation because of sample variance or minor variations in their interpretation of the protocol or method.

**Reviewer Confidence:**

4: Quite sure. I tried to check the important points carefully. It's unlikely, though conceivable, that I missed something that should affect my ratings.

---

> ### Author Rebuttal · Authors · 2023-08-28
>
> Thank you for your valuable feedback and suggestions. For your raised concerns, we will address them one by one as shown in the following:
>
> **Reasons to Reject:**
>
> 1. We appreciate the comment and we also acknowledge the great number of sources is valuable as discussed in Limitation. However, we would like to clarify that this work focuses more on **the planning capability of LLMs to decide whether or not require knowledge, which source to call, and when to call.** Specifically, we target the interaction and dependency issues between different sources of multi-source knowledge-grounded dialogue generation. Distinguishing from previous works that incorporate all sources indiscriminately (resulting in cumbersome processes and unnecessary computing cost), we explore three different situations: response without requiring knowledge; response requires one source of knowledge; and response requires two dependent resources of knowledge.
>
> 2. Our dataset KBP is constructed targeting personalized response generation, which is **one of the key research problems in dialogue tasks**. We utilize knowledge related to persona to investigate dependency issues across different sources of knowledge. As shown in Figure 1(a) and Appendix A, without the assistance of related knowledge, it is relatively easy to generate persona-inconsistent responses.
>
> 3. We provide both supervised methods and unsupervised methods to investigate the performance of LLMs as source planners. For the supervised methods, the LLMs are trained to learn the planning capability and grounding capability simultaneously. Since the planning step is based on the dialogue context, it is agnostic to the knowledge used in the dataset. We also want to note that there is no other suitable dataset (e.g., contains responses that do not use knowledge and use multiple sources of knowledge simultaneously; and the dependency between different sources of knowledge), and the persona and documents in the test set of our proposed KBP are unseen during the training.
>
>
> 4. We have considered some baselines from previous works such as FoCus [1] and MSKE-Dialog [2] which target multi-source knowledge-grounded dialogue generations. However, we found that almost all previous works assumed that the response must require external knowledge and incorporate all sources indiscriminately. For example, FoCus assumed that each response must require knowledge. We focus on more complex and frequent situations in practice where the response may need or not need the involvement of different sources of knowledge. Furthermore, we also model the dependency relationship between different sources. Despite handling different problem settings, we further conduct experiments with these methods on the concerned problem below (Please refer to Question D).
>
> **Questions for The Authors:**
>
> A: Lines 302-307 introduce the dependency issues across two different sources of knowledge. In our paper, the DOCUMENTS knowledge is reliant on PERSONA knowledge, as shown in Figure 1(b), There are many pieces of knowledge (for example, commonsense knowledge) behind one persona description. For simplicity, we need to open the Music App to find the latest songs. Here we use "I love to listen to pop music" as a persona description, and some names of pop songs as knowledge such as "current pop songs include xxx". When the user asks "Which song is your favorite ?", we find the persona description first and then check the song list, which is exactly like opening the Music App to find the playlist. The existence of this dependency requires us to design a framework to capture the relationships between different sources and decide the call order sequences. **There are two steps that need to be carefully designed: planning and retrieval.** The planning decides the order of sources, once you decide the wrong order, there is no chance to get the correct answer. The retrieval step needs to be executed according to the outputs of planning. We provide additional analysis of the dependency issue in Tables 5 and 7.
>
> B. Thanks for the insightful question. Yes, our framework can be easily extended to encompass additional knowledge sources. Despite our main focus not incorporating more and more knowledge sources, we indeed evaluate the performance of our method in real-world applications by incorporating Wikipedia knowledge or other types of knowledge in a similar setting (dependency and multiple sources). The extent of its effectiveness in specific tasks can vary, and fine-tuning might be necessary to achieve optimal performance for certain applications such as many sources of knowledge are required. Generally speaking, it performs well at planning and assembling different sources of knowledge, especially after in-domain finetuning (user satisfaction improves from 67% to 82%).
>
> C: We mainly follow the method proposed in this paper [3]. For your second question, the answer is yes. The persona config is fixed for each dialogue in our experiments. Since our method is agnostic with the config, it can be modified and updated as the conversation goes on [4].
>
> D: We provide the results in Table 1 by using the method proposed in the FoCus [1] and other methods target multi-sources knowledge-grounding dialogue generation can not be directly applied to KBP (if there is, pls let us know), and we use two Chinese dialogue models CDialGPT [5] and Pangu-Bot [6] as backbones for comparison. We can find that our method performs best and we want to note the FoCus method assumes all responses require knowledge (and the problem setting is also different) thus the comparison may not be fair here (as explained in the response above).
>
> Table 1: The performance between the method proposed in FoCus and SAFARI.
> | Model                      | BLEU1 | RougeL | P.C   | K.C   |
> |----------------------------|-------|--------|-------|-------|
> | CDialGPT + PG + KG   | 22.51 | 25.29  | 64.52 | 17.90 |
> | Pangubot + PG + KG   | 18.06 | 20.23  | 67.20 | 10.41 |
> | ChatGLM (ours) | 23.81 | 26.70  | **76.99** | 42.39 |
> | BELLE (ours)   | **30.48** | **34.61**  | 75.34 | **46.62** |
>
> [1] Call for customized conversation customized conversation grounding persona and knowledge. AAAI 2022
>
> [2] More is Better: Enhancing Open-Domain Dialogue Generation via Multi-Source Heterogeneous Knowledge. EMNLP 2021
>
> [3] Long time no see open-domain conversation with long-term persona memory. ACL 2022
>
> [4] Prompted LLMs as Chatbot Modules for Long Open-domain Conversation. ACL 2023
>
> [5] A Large-Scale Chinese Short-Text Conversation Dataset
>
> [6] PanGu-Bot: Efficient Generative Dialogue Pre-training from Pre-trained Language Model

---

### Meta-Review · Area_Chair_Rf8d · 2023-09-19

**Recommendation:** 3

**Metareview:**

The paper presents SAFARI as a framework to leverage the LLM's planning capability to incorporate multiple knowledge sources for personalized knowledge-grounded dialogue response generation. The paper also introduces a new dataset KBP for experiments, and experiments shows the SAFARI is effective at producing persona-consistent and knowledge enhanced dialogue responses.

While the reviewers generally agreed the paper is well-written and easy to follow, the knowledge within the KBP dataset is limited to persona-related information, making it less clear whether the SAFARI framework can generalize towards general knowledge-grounded dialogues.

---

### Decision · Program_Chairs · 2023-10-07

**Decision:**

Accept-Findings

**Comment:**

The paper presents SAFARI as a framework to leverage the LLM's planning capability to incorporate multiple knowledge sources for personalized knowledge-grounded dialogue response generation. The paper also introduces a new dataset KBP for experiments, and experiments shows the SAFARI is effective at producing persona-consistent and knowledge enhanced dialogue responses.

While the reviewers generally agreed the paper is well-written and easy to follow, the knowledge within the KBP dataset is limited to persona-related information, making it less clear whether the SAFARI framework can generalize towards general knowledge-grounded dialogues.